# Tripartite-Motif 21 (TRIM21) Deficiency Results in a Modest Loss of Herpes Simplex Virus (HSV)-1 Surveillance in the Trigeminal Ganglia Following Cornea Infection

**DOI:** 10.3390/v14030589

**Published:** 2022-03-12

**Authors:** Amanda Berube, Grzegorz B. Gmyrek, Derek J. Royer, Daniel J. J. Carr

**Affiliations:** 1Department of Ophthalmology, The University of Oklahoma Health Sciences Center, Oklahoma City, OK 73104, USA; amanda-berube@ouhsc.edu (A.B.); grzegorz-gmyrek@ouhsc.edu (G.B.G.); 2Department of Biology, East Texas Baptist University, Marshall, TX 75670, USA; droyer@etbu.edu; 3Department of Microbiology and Immunology, The University of Oklahoma Health Sciences Center, Oklahoma City, OK 73104, USA

**Keywords:** HSV-1, cornea, trigeminal ganglia, T cells, interferon-γ, interferon stimulatory genes, TRIM21

## Abstract

Tripartite-motif 21 (TRIM21) is thought to regulate the type I interferon (IFN) response to virus pathogens and serve as a cytosolic Fc receptor for immunoglobulin. Since herpes simplex virus (HSV)-1 is sensitive to type I IFN and neutralizing antibody, we investigated the role of TRIM21 in response to ocular HSV-1 infection in mice. In comparison to wild type (WT) mice, TRIM21 deficient (TRIM21 KO) mice were found to be no more susceptible to ocular HSV-1 infection than WT animals, in terms of infectious virus recovered in the cornea. Similar pathology, in terms of neovascularization, opacity, and loss of peripheral vision function, was observed in both WT and TRIM21 KO mice. However, TRIM21 KO mice did possess a significant increase in infectious virus recovered in the trigeminal ganglia, in comparison to the WT animals. The increased susceptibility was not due to changes in HSV-1-specific CD4+ or CD8+ T cell numbers or functional capabilities, or in changes in type I IFN or IFN-inducible gene expression. In summary, the absence of TRIM21 results in a modest, but significant, increase in HSV-1 titers recovered from the TG of TRIM21 KO mice during acute infection, by a mechanism yet to be determined.

## 1. Introduction

Herpes simplex virus (HSV)-1 is a double-stranded DNA virus that elicits a robust inflammatory response following ocular infection in the naive mouse, often resulting in significant tissue pathology [1,2,3]. While the incidence of HSV-1 infection in the human host overall is declining in industrialized countries, this pathogen continues to plague mankind and is now the leading cause of genital herpes infection in the United States and elsewhere [4,5]. Responsible for the success of the pathogen are the countermeasures elicited by virus-encoded proteins that facilitate immune evasion or suppression. For example, HSV-1-encoded glycoprotein (g)C, gE, and gI interfere with various aspects of antibody and complement action, including antibody-dependent cell cytotoxicity [6,7,8], whereas HSV-1-infected cell protein (ICP)47 blocks major histocompatibility complex (MHC) class I presentation of virus peptides to CD8+ cytotoxic T cells [9]. The virus also disrupts T cell receptor signaling at the linker for activation of T cell stage [10] and suppresses the invariant chain associated with MHC class II processing; thus, reducing MHC class II antigen presentation that is important in CD4+ T cell activation [11]. HSV-1, not only targets processes associated with the adaptive immune response, but also interferes with a number of anti-viral pathways associated with the innate immune response [12], including oligoadenylate synthetase/RNase L and STING/p204/IFI16 pathways, thought to be important in controlling HSV-1 replication in the cornea of mice [13,14]. HSV-1 also evades immune detection, establishing a ‘latent’ infection in neurons, where MHC class I molecules remain quiescent, even though the virus undergoes abortive reactivation with ongoing transcript expression in latently infected neurons [15,16]. The overall success of an HSV-1 infection depends upon the status of the naive host relative to the type I IFN response [3], which is influenced by the site of initial infection, as well as the likelihood of a sufficient (duration) and broad antigen exposure, to achieve significant coverage by the adaptive immune system against virus-encoded products.

Numerous cytosolic sensors detect viral RNA or DNA, which leads to the activation of type I IFNs [17,18,19,20]. Whereas type I IFNs are potent anti-viral compounds, chronic or excessive production of these cytokines has been associated with autoimmune disease and functional neuropathology [21,22,23]. Therefore, it is paramount to control the expression of IFN, once the resolution that prompts IFN expression occurs. The suppressor of cytokine signaling (SOCS) molecules, SOCS1 and SOCS3, have been found to inhibit Janus kinase signaling and, thus, dampen IFN expression [24]. Furthermore, SOCS1 and SOCS3 are induced by HSV-1, which would favor replication and spread of the virus in the host [25,26]. Another family of molecules that are upregulated by type I IFN and thought to influence downstream pathways activated by type I IFN are the tripartite-motif (TRIM) proteins [27]. The N-terminal domain of this family of proteins is known to possess E3 ligase activity, which targets ubiquitin and interferon stimulated protein of 15 kDa [28]. One specific TRIM protein, TRIM21, has been described as advantageous or deleterious to host resistance to virus infection based on targeting interferon regulator factor (IRF)3 [29,30]. In addition to the regulation of IFN downstream pathways, TRIM21 has also been found to be a cytosolic Fc receptor for IgG, IgM, and IgA molecules, which can then neutralize and degrade antibody-bound targets, including viruses [31,32].

In the present study, we investigated the role of TRIM21 in host resistance to ocular HSV-1 infection using TRIM21 deficient (TRMI21 KO), heterozygous (TRIM21+/−), and wild type (WT) mice, assessing virus replication and spread, the immune response, and cornea pathology parameters, to characterize and compare predicted changes between WT and TRIM21 KO mice. The results showed no difference in virus titer, inflammation, or pathology in the cornea of infected mice. However, there was a noticeable difference in virus replication, with 10-fold more infectious virus recovered from the trigeminal ganglia (TG) of TRIM21 KO mice compared to WT animals. This observation was consistent with an increase in lytic gene expression in the TG of TRIM21 KO mice compared to WT or TRIM21+/− animals. The increase in virus replication was not due to changes in the function of infiltrating cells, as CD4+ and CD8+ T cells isolated from the TG of infected mice all showed similar levels of IFN-γ expression post stimulation. Likewise, type I IFN and IFN-inducible gene expression levels within the TG were consistent between mouse genotypes. Therefore, the absence of TRIM21 expression specifically impacts the host control of HSV-1 replication in the TG by mechanisms yet to be determined

## 2. Materials and Methods

### 2.1. Mice and Infection

A breeder pair of heterozygous TRIM21 mice were obtained from Jackson Laboratory (Bar Harbor, ME, USA) and used to establish a colony. Offspring were genotyped to identify homozygous wild type (WT), heterozygous TRIM21 (TRIM21+/−), and deficient TRIM21 (TRIM21 KO) male and female mice, which were subsequently used in all experiments. Five- to six-week old mice were anesthetized with xylazine (5.0 mg/kg) and ketamine (100 mg/kg) administered intraperitoneally and then infected with HSV-1 (1000 plaque forming units (PFU/cornea) in 3 µL of phosphate buffered saline (PBS, pH = 7.4), following scarification of the cornea using a 25-gauge needle. All animals were housed in a specific pathogen-free vivarium at the Dean McGee Eye Institute on the University of Oklahoma Health Sciences Center campus, under the approved institutional animal use and care committee protocol, 19-060-ACHIX. At the indicated time, mice were exsanguinated, as previously described [33].

### 2.2. Virus and Plaque Assay

HSV-1 (strain McKrae) was propagated in the Vero African green monkey cell line (American Type Culture Collection, Manassas, VA, USA) with a stock (1.2 × 10^8^ PFU/mL) stored at −80 °C until use. Virus titers in the cornea and TG were determined by plaque assay on day 3 and 7 post infection (pi), as previously described [34]. 

### 2.3. Real Time Reverse Transcriptase (RT)-Polymerase Chain Reaction (PCR)

The cornea and WT mice were collected at the indicated days pi or in uninfected WT mice. RNA was isolated using Trizol (Thermofisher, Waltham, MA, USA), and cDNA was generated using an iSCRIPT cDNA synthesis kit (Bio-Rad, Hercules, CA, USA). Forward and reverse oligonucleotide primers were used to amplify the targeted genes of interest by RT-PCR, as noted below Table 1:

A proprietary set of forward and reverse primers were obtained from a commercial vendor (BioRad) targeting GAPDH, TRIM21, IFNα1, IFNβ, and Bst2 to amplify targeted genes. The oligonucleotide primers are validated by the vendor for specificity and primer/dimer formation. Relative values of gene expression were determined via ∆∆Ct method, using GAPDH to normalize samples [35]. Amplification of targeted gene using the gene amplicon format was as follows: 220 ng cDNA of sample was initially activated at 95 °C for 30 s, followed by a denaturation (95 °C for 5 s) and a annealing/extension step (60 °C for 30 s) repeated 40 times and a final hold step at 4 °C. A SYBER green ITaq Supermix (Bio-Rad) was used as part of the amplification, to detect amplified product. Melt curves (65–95 °C, 0.5 °C increments with 5 s/step) were used to establish a single product amplification. A CFX Connects thermal cycler (Bio-Rad) was used for all PCR experiments to monitor target amplification, with CFX Manager software to analyze the data. Threshold cycles were determined after subtracting the background fluorescence of each sample, where the relative light units achieved a level 10 standard deviations above the baseline relative light units. 

### 2.4. Spectral Domain-Optical Coherence Tomography (SD-OCT)

Prior to and following infection, the corneas of mice were imaged using a Bioptigen SD-OCT system (Durham, NC, USA) for corneal thickness. Images were acquired using Bioptigen SD-OCT software (InVivoVue Clinic, Bioptigen, Morrisville, NC, USA).

### 2.5. Corneal Sensitivity

To measure the mechanosensory function of infected mice, a Cochet Bonnet esthesiometer (Luneau SAS, Rue Roger Bonnet, France) was employed. Briefly, infected mice, at the indicated time, were held firmly behind their neck and a monofilament varying in length from 0.5 to 6.0 cm was touched perpendicular to the cornea divided into four quadrants, and the length of the filament to elicit a blink response was recorded for each eye. The genotype of the infected mice was not apparent to the examiner until after the test. Uninfected mice served as the control to establish baseline readings.

### 2.6. Cornea Pathology

The opacity of the cornea was determined at 0, 7, and 30 days pi, following the removal of the cornea button from exsanguinated mice and assaying the tissue for absorbance of 500 nm using FLUOstar Omega plate reader (BMG Labtech, Offenburg, Germany), as described previously [33]. Following the measurement, the cornea was fixed using 4% paraformaldehyde (Sigma-Aldrich, St. Louis, MO, USA) for 30 min and washed in PBS containing 1% Triton X-100 (Sigma Aldrich). Following an incubation overnight with PBS containing 10% donkey serum (Abcam, Boston, MA, USA), the tissue was labeled for blood (CD31+) and lymphatic (LYVE1+) vessels, as previously described [36]. Alternatively, corneas were stained with rabbit anti-Trim21 (Abcam) and FITC-conjugated goat anti-HSV-1 antibody (Abcam) at a 1:1000 dilution. Following an incubation overnight with PBS containing 10% donkey serum (Abcam), the tissue was stained with donkey anti-rabbit Alexa Fluor 568 (Invitrogen, Eugene, OR, USA) at a 1:1000 dilution. Following an additional washing step, incisions were made into the cornea into quarters, such that the cornea could be placed on a slide for whole mount imaging in Dapi mounting media containing 50% glycerol. An Olympus FV1200 scanning confocal microscope (Center Valley, PA, USA) was used to sequentially scan labeled corneas and generate z-stacked images. The total area positive for vessels per field of view (4 quadrants/cornea) was analyzed and quantified using Metamorph software (Molecular Devices Inc., San Jose, CA, USA).

### 2.7. Flow Cytometry

TG from exsanguinated WT, TRIM21+/− and TRIM21 KO mice were collected at day 7 pi. Single-cell suspensions were generated using a 2 mL Wheatley Dounce homogenizer (Fisher Scientific, Waltham, MA, USA) in Dulbecco’s modified eagle medium (DMEM) containing high glucose, L-glutamine, and pyruvate (Gibco Life Technologies, Grand Island, NY, USA) containing 10% fetal bovine serum (FBS, Gibco Life Technologies) and antibiotic/antimycotic solution (Thermofisher), and passed through a 70-µm cell strainer (Cell Treat Scientific Products, Pepperell, MA, USA). Cells were then washed in PBS (supplemented with 2% FBS; herein referred to as staining buffer, SB) and finally resuspending in the same buffer. Next, the cell suspension was stained with amine-reactive fluorescent dye Zombie Aqua (Biolegend, San Diego, CA, USA), used to assess live vs. dead status of TG cells. Non-specific binding sites of TG single-cell suspensions were blocked with 800 ng of CD16/CD32 (Catalog # 16-0161-82, clone 93, EBioscience, San Diego, CA, USA) for 15 min at 4 °C. The cells were then labeled with a combination of anti-CD45 (clone 30-F11) conjugated with Pacific Blue, (Catalog # 103126), -CD3 (clone 17A2) conjugated with PE-Cy7 (Catalog # 100220), -CD4 (clone GK1.5) conjugated with APC (Catalog #100412), and -CD8 (clone 53-6.7) conjugated with APC-Cy7 (Catalog #100714) (all from BioLegend, San Diego, CA, USA) at 800 ng/antibody/sample. To identify HSV-1-specific CD8+ T cells, cells were initially stained with 500 ng of one of the following PE-or Alexa Fluor 488-conjugated tetramers: H-2K(b)/SSIEFARL (glycoprotein B, gB), H-2K(b)/INNTFLHL (infected cell protein [ICP]8), or H-2K(b)/QTFDFGRL (ICP6) (all from NIH Tetramer Core Facility, Emory University, Atlanta, GA, USA), and incubated for 30 min on ice. Next, the cells were washed with PBS/2% FBS and fixed with 2% paraformaldehyde. To determine the number of HSV-1 -specific CD4 T cells, the cell suspension was stained with a combination of anti-CD45, -CD3, -CD4, and -CD8, along with BV421-conjugated tetramer I-A(b) IPPNWHIPSIQDA (glycoprotein D, gD, from NIH Tetramer Core Facility) for 1 h at 37 °C. Subsequently, the cells were washed with PBS/2% FBS and fixed with 2% paraformaldehyde until acquisition 24–48 h later using a spectral flow cytometer Aurora (Cytek Biosciences, Fremont, CA, USA) and analyzed using FlowJo software (FlowJo LLC, Medford, OR, USA).

For evaluation of interferon gamma (IFN-γ) producing T cells, the single TG cell suspension was placed into RPMI (supplemented with 10 mM D-glucose, 10% FBS, 1mM sodium pyruvate 2 mM L glutamine, and 1× antibiotic-antimycotic solution) and stimulated with 100 nM phorbol 12-myristate 13-acetate (PMA) and 1 μM ionomycin (both from MilliporeSigma, Burlington, MA, USA) for 6 h in the presence of brefeldin A (GolgiPlug from BD Biosciences/ThermoFisher Scientific ). After the incubation period, the cells were washed with SB and stained with Zombie Aqua, followed by washing and staining with antibody cocktail (consisting of anti-CD45, CD3, CD4, or CD8) and PE-, AlexaFluor488, or BV421-conjugated HSV-1 tetramers, as described above, with the same incubation conditions. Finally, the cells were fixed and permeabilized with Foxp3/Transcription Factor Staining Buffer Set (Thermo Fischer Scientific) according to the manufacturer recommendations. At the permeabilization step, anti-mouse IFN-γ Ab conjugated with FITC (Thermo Fischer Scientific) was added to the cell suspension, followed by incubation at room temperature for 45 min. Subsequently, the cells were washed with permeabilization buffer, acquired using a spectral flow cytometer Aurora (Cytek Biosciences), and analyzed using FlowJo software (FlowJo LLC).

### 2.8. Statistical Analysis

Data were analyzed for significance using (*p* < 0.05) Prism 9.0 software (GraphPad, San Diego, CA, USA), with tests used to determine the significance indicated in each figure legend.

## 3. Results

### 3.1. HSV-1 Infection Induces TRIM21 Expression in the Cornea

TRIM21 expression has previously been reported to be upregulated in the cornea of mice in response to HSV-1 infection [37]. To confirm this observation in our model, we initially evaluated TRIM21 gene expression over time post HSV-1 infection. Whereas, there was little detection of TRIM21 mRNA expression in the uninfected cornea, TRIM21 levels dramatically rose 2 days pi and maintained that level through day 7 pi (Figure 1A). However, by day 14 pi TRIM21 mRNA expression had dropped back down to baseline levels, suggesting a dynamic but temporal TRIM21 gene expression response in the cornea following HSV-1 infection. Similar to the cornea, TRIM21 mRNA was not readily detected in the TG of uninfected mice (Figure 1B). However, by day 2 pi, TRIM21 expression significantly rose and was maintained throughout the period of surveillance (day 14 pi), following HSV-1 infection (Figure 1B). We also investigated TRIM21 expression at the protein level and found that, consistent with mRNA expression, TRIM21 protein (red) was not detected in the uninfected cornea (Figure 1C) but was upregulated in HSV-1-infected cornea, expressed by cells that were (yellow) and were not infected with HSV-1 (Figure 1D). It should be noted that a large number of HSV-1-infected cells (green) did not express TRIM21 (Figure 1D). Taken together, HSV-1 elicits a significant upregulation of TRIM21 expression in infected tissue, which is temporal or chronic dependent on the tissue.

### 3.2. The Absence of TRIM21 Is Reflected in a Loss of Virus Surveillance in the TG, but Not in Cornea of HSV-1-Infected Mice

TRIM21 has previously been reported to contribute to host defense against lethal mouse adenovirus-1 infection following intraperitoneal administration [38], likely through a cGAS-dependent mechanism [39]. However, in the case of HSV-1 infection, targeting TRIM21 expression using siRNA silencing, it was reported that a loss of TRIM21 expression resulted in a decrease in HSV-1 shedding in the tear film during acute infection [37]. Since we had found that virus shedding does not necessarily correlate with HSV-1 tissue levels [40], virus titers were obtained from the cornea and TG at times pi in WT, TRIM21+/−, and TRIM21 KO mice. At 3 days pi, there was no difference in the amount of infectious virus recovered from the cornea (Figure 2A) or TG (Figure 2B) of WT, TRIM21+/−, and TRIM21 KO mice. Likewise, infectious virus titers obtained from the cornea of WT, TRIM21+/−, and TRIM21 KO mice were all similar in quantity at day 7 pi (Figure 2C). However, infectious HSV-1 levels were significantly higher in the TG of TRIM21 KO mice compared to the WT animals at day 7 pi (Figure 2D).

### 3.3. The Loss of TRIM21 Does Not Modify Corneal Pathology Compared to WT Mice

Acute corneal infection of mice results in tissue pathology, including opacity and neovascularization [36,41,42,43]. Therefore, corneal opacity was quantified during acute infection (day 7 pi) into latency (day 30 pi). The results show that, similar to WT and TRIM21+/− mice, TRIM21 KO mice displayed the same degree of opacity at day 7 (Figure 3A) and day 30 (Figure 3B) pi. The opacity increased as the infection progressed from day 7 to day 30 pi, with significant differences comparing the TRIM21+/− and TRIM21 KO at day 7 to day 30 pi.

Consistent with these results, corneal neovascularization was also similar between genotypes, with WT, TRIM21+/− and TRIM21 KO showing the same degree of blood and lymphatic vessel genesis into the central cornea by day 30 pi (Figure 4). Representative corneal vascularization images for WT, TRIM21+/−, and TRIM21 KO show similar levels of blood and lymphatic vessel genesis into the central cornea (Figure 4A), summarized for area occupied by blood (Figure 4B) and lymphatic (Figure 4C) vessels at 30 days pi.

HSV-1 infection is also known to elicit cornea edema, likely due to the expression of pro-inflammatory cytokines and recruitment of leukocytes, including neutrophils, via chemokines to the site of infection [44,45,46]. We previously utilized SD-OCT to assess corneal edema following HSV-1 infection of mice [47]. Therefore, we assessed whether the absence of TRIM21 impacted the level of cornea edema, following HSV-1 infection. WT, TRIM21+/−, and TRIM21 KO mice were infected with HSV-1 and evaluated for cornea edema over time, until day 30 pi. Prior to infection, mice were imaged by SD-OCT, to establish a baseline. The results showed that, although HSV-1 induced a 30–40% increase in cornea edema, there was no difference in the amount of swelling between the genotypes of animals tested (Figure 5A,B). Taken together, the absence of TRIM21 within the cornea was not reflected by quantitative changes in ocular pathology, suggesting that TRIM21 plays little, if any, role in the host response to local virus infection and tissue pathology within the cornea.

### 3.4. Visual Axis Function Is Reduced in WT and TRIM21 KO Mice Following HSV-1 Infectionn

The consequences of ocular HSV-1 infection in mice often results in the disruption of functional parameters associated with vision or the integrity of the visual axis. Specifically, HSV-1 infection of the cornea often leads to a loss in mechanosensory function from denervation, including a loss of substance P-expressing sensory fibers, which are thought to contribute to wound healing [48,49,50,51,52]. One means of measuring mechanosensory function of the cornea is the blink reflex. Using a Cochet Bonnet esthesiometer, HSV-1-infected WT, TRIM21+/−, and TRIM21 KO mice were evaluated for cornea sensitivity. All groups of infected mice showed a significant loss in the blink response at day 7 pi (Figure 6A). WT mice tended to recover by day 30 pi, whereas TRIM21+/− and TRIM21 KO mice still showed a 30% loss in function. However, due to variability between measurements within the same group of TRIM21+/− and TRIM21 KO animals, the loss at 30 days pi was not significant.

The perception of optomotor movement by rodents, including mice, can be assessed using a virtual optokinetic tracking system [53]. To determine if changes in visual acuity occurred in WT, TRIM21+/− and TRIM21 KO mice following HSV-1 infection, we measured head movement under binocular viewing conditions and found that all groups of mice lost vision by day 15 pi, with a permanent loss of 25% measured at day 30 pi (Figure 6B). Thus, as in the tissue pathology results, the function of the visual axis appeared to be similarly compromised in all groups of mice evaluated post HSV-1 infection.

### 3.5. T Cell Infiltration and Function Are Similar between WT, TRIM21+/−, and TRIM21 KO Mice

Historically, T cells are thought to control HSV-1 replication in the TG during acute and latent infection following ocular HSV-1 challenge [54,55,56,57]. Since infectious virus detected in the TG of TRIM21 KO mice was equivalent to that found in the WT mice at day 3 pi, but elevated by day 7 pi, we interpreted these results as suggesting that the adaptive arm of the immune system likely plays a more significant role compared to the innate immune response within the TG of TRIM21 KO mice, relative to virus surveillance. Therefore, we initially investigated the T cell infiltrate and function in the TG at day 7 pi in WT, TRIM21+/−, and TRIM21 KO mice. The results showed no deficiency in the number of total (Figure 7A,C) or HSV-1-specific CD4+ or CD8+ T cells (Figure 7B,C) residing in the TG of HSV-1-infected TRIM21 KO mice compared to WT or TRIM21+/− animals.

Moreover, the function of HSV-1-specific CD4+ and CD8+ effector T cells, as measured by IFN-γ expression post-stimulation, did not reveal any significant differences in T cells from the WT, TRIM21+/−, and TRIM21 KO moue TG (Figure 8A,B). We interpreted these results as suggesting that T cell mobilization and function were not compromised in the TRIM21 KO mice, which would not explain the increase in virus titer captured in the TG of these animals at day 7 pi.

### 3.6. TG of TRIM21 KO Mice Display an Elevation in HSV-1 Lytic Gene Expression but Not Type I IFN or IFN-Inducible Genes Following Infection

Since greater susceptibility of TRIM21 KO mice to HSV-1 replication in the TG could not be explained by a deficiency in T cell infiltrate or function, other mechanisms were explored. TRIM21 is thought to be regulated by IFN regulatory elements, including IFN regulatory factor (IRF)1 and STING, as well as regulate type I IFN expression through IRF3 [37,58,59,60]. Moreover, type I IFNs and STING-mediated pathways contribute to the host defense against HSV-1 infection, by preventing replication of the virus [14,19,61,62,63]. Therefore, we, next evaluated levels of type I IFN and downstream effector pathways activated in response to type I IFN in the TG of HSV-1-infected mice. The results showed that the levels of type I IFNs, including IFN-α1, IFN-α4, and IFN-β, which have previously been found to be potent inhibitors of HSV-1 replication [35], were similar between TG from WT, TRIM21+/−, and TRIM21 KO mice (Figure 9A). Furthermore, downstream effector molecules of type I IFN-activated pathways that antagonize HSV-1, including tetherin (*Bst2*) [64] and oligoadenylate synthetases (OAS) [13,65] were found to be expressed at equivalent levels, as was STING and DAI in the TG of HSV-1-infected WT, TRIM21+/− and TRIM21 KO mice (Figure 9A), even though HSV-1 lytic gene expression (including ICP27, TK, and gB) was elevated in the TG of TRIM21 KO mice (Figure 9B). Therefore, the increased susceptibility of TRIM21 KO mice to HSV-1 replication in the TG, following ocular infection, is not due to changes in type I IFN expression or changes in the levels of IFN-inducible effector molecules OAS1a or tetherin found in the tissue.

## 4. Discussion

In the present study, we found local HSV-1-induced TRIM21 corneal expression during acute infection is temporal in nature returning to baseline levels by day 14 pi. However, TRIM21 does not appear to play a local role in host resistance against ocular HSV-1 challenge or virus-induced pathology, as there were no significant differences when comparing viral titer or virus-induced pathology in the cornea of WT to TRIM21 KO mice. In contrast to these results, the absence of TRIM21 was reflected by an elevation in the infectious virus content and virus lytic gene expression found in the TG by day 7 pi. As in the cornea, TRIM21 expression was significantly elevated in the TG following cornea infection. Unlike the cornea, TRIM21 expression did not dissipate in the TG following the clearance of infectious virus at day 14 pi. Since T cells are thought to contribute to virus surveillance in the TG during acute infection [56], we investigated the level and function of CD4+ and CD8+ T cells at the time when the virus load was elevated in the TRIM21 KO animals. We found there was no difference in the number of CD4+ or CD8+ T cells comparing WT, TRIM21+/−, and TRIM21 KO mice. Similarly to the incidence of T cell populations found in the HSV-1-infected TG, the number of IFN-γ-expressing CD4+ or CD8+ T cells was consistent amongst the WT, TRIM21+/−, and TRIM21 mice. While it is possible the location of effector T cells may not reside in close proximity to HSV-1-infected cells within the TG, to significantly influence virus replication, we surmise that it is more likely (an)other mechanism(s) may explain the loss in resistance to HSV-1 replication in the TG of TRIM21 KO mice.

TRIM21 is thought to regulate pathways associated with type I IFN response during virus infection. In human microglia, an increase in TRIM21 expression is thought to target and prevent IRF3 phosphorylation, resulting in the suppression IFN-β production in response to the single-stranded RNA virus, Japanese encephalitis virus [59]. Additional results were reported investigating the role of TRIM21 expression in mice, which showed TRIM21 KO mice yielded significantly less HSV-1 compared to WT controls, following intraperitoneal infection [66]. In this model, the negative effect of TRIM21 expression on control of HSV-1 infection was thought to be due to degradation of the cytosolic DNA sensor, DDX41, resulting in a loss of IFN-β production [66]. However, another study found TRIM21 stabilizes IRF3 expression resulting in an increase in resistance to virus infection; in this case, against Sendai virus [30]. As TRIM21 expression is tightly regulated by IRF pathways and the end-products, IFN-α and IFN-β, as well as IFN-γ [58], and as these cytokines are produced in response to HSV-1 infection [61,67,68], we investigated whether increased susceptibility to infection was reflected by changes in IFN levels or IFN-inducible gene expression. Even though there was an elevation in the HSV-1 lytic gene expression from TG of TRIM21 KO mice, we did not find any difference in IFN or IFN-inducible gene expression comparing the TRIM21 KO to WT or TRIM21+/− animals. Since the collection of tissue was a single time point, which coincided with when viral titers were assessed, it is possible earlier time points may have been more revealing. Alternatively, other anti-viral mechanisms, including apoptosis [69,70] or necroptosis [71], may contribute to host defense against HSV-1 infection in the TG, as TRIM21 is associated with the tumor necrosis factor-related apoptosis-inducing ligand (TRAIL)-induced necrosome [72] and, thus, may prevent the use of elements within the cell that facilitate virus replication. At this point, we surmise that the microenvironment, which can include infiltrating myeloid-derived cells and NK cells, as well as the virus pathogen, greatly influences the positive or negative regulation of TRIM21 expression, which ultimately exerts influence on host resistance to virus infection, in this case, HSV-1.

It should be noted that the present study modeled acute infection into latency of naive mice. However, we did not evaluate latent virus content in our model and, therefore, cannot comment on whether the absence of TRIM21 influences the establishment of HSV-1 latency. It is possible that in the context of a prophylactic vaccine or upon repeat infection with a non-persistent pathogen, the Ig-Fc function of TRIM21 may be instrumental in the outcome of host resistance.

## Figures and Tables

**Figure 1 viruses-14-00589-f001:**
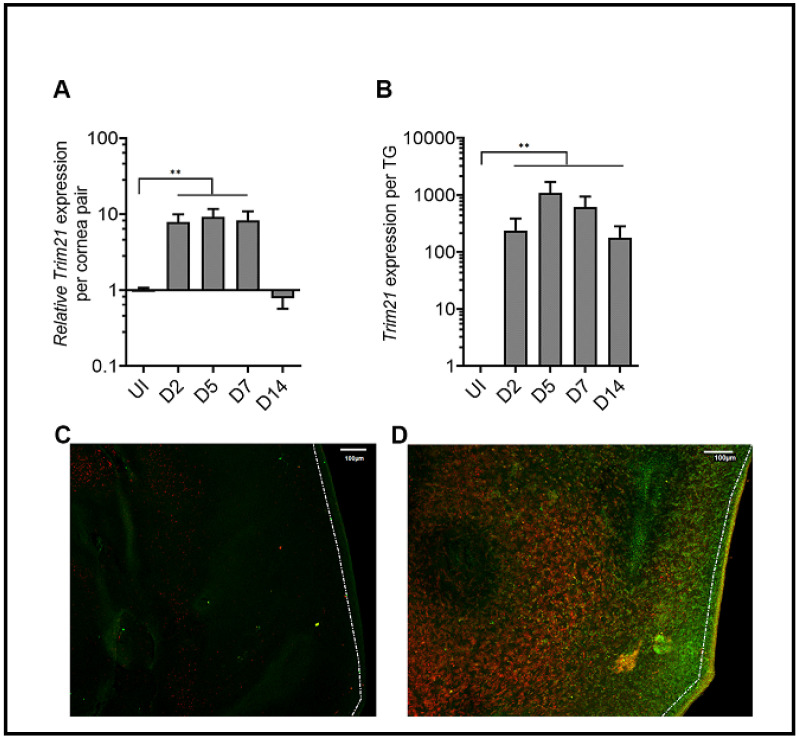
TRIM21 expression in the cornea and trigeminal ganglia. Male and female C57BL/6 (B6) mice were infected in the cornea with HSV-1 (10^3^ plaque forming units (PFU) in 3 ul PBS. At the indicated time post infection (pi), the (**A**) corneas (5–7 samples/time point) or (**B**) trigeminal ganglia (TG) (3–5 samples/time point) of infected mice were removed and assessed for expression of TRIM21 by real-time RT-PCR. Bars represent the mean relative value + SEM, ** *p* < 0.01, comparing the indicated groups as determined by one-way ANOVA and Kruskal–Wallis multiple comparison test. Representative confocal images of (**C**) uninfected and (**D**) infected corneas stained for TRIM21 (red) and HSV-1 (green) antigen at 3 days post infection. Uninfected (UI) mice served as controls. The white scale bar = 100 µm.

**Figure 2 viruses-14-00589-f002:**
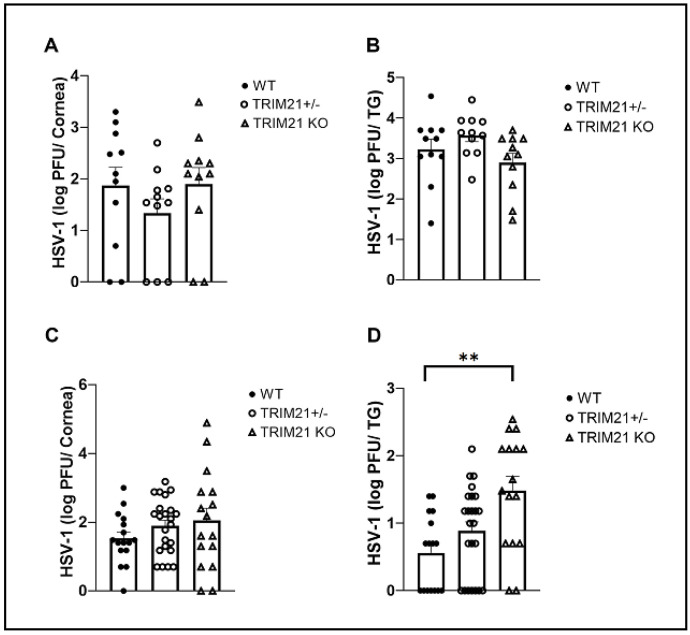
HSV-1 levels are elevated in the trigeminal ganglia of TRIM21 KO mice at day 7 post infection. Male and female C57BL/6 (WT), TRIM21+/−, and TRIM21 KO mice (n = 11–22/group) were infected with HSV-1 (10^3^ PFU/cornea). At day 3 (**A**,**B**) or day 7 (**C**,**D**), the corneas (**A**,**C**) and trigeminal ganglia (TG) (**B**,**D**) of infected mice were removed and processed to determine viral load/tissue by standard plaque assay. Bars represent mean + SEM, ** *p* < 0.01 comparing the TRIM21 KO to WT mice day 7 pi, as determined by one-way ANOVA and Kruskal–Wallis multiple comparison test.

**Figure 3 viruses-14-00589-f003:**
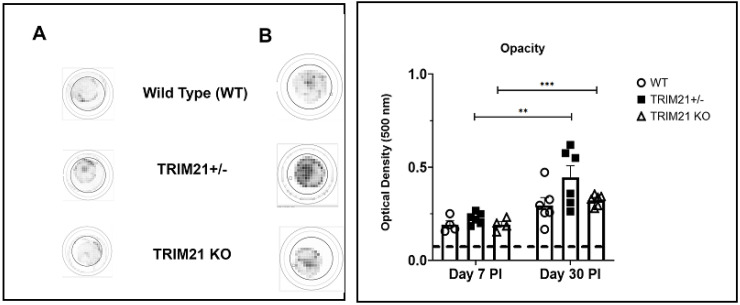
Corneal opacity is equivalent amongst WT, TRIM21+/−, and TRIM21 KO mice at different times post HSV-1 infection. Male and female C57BL/6 (WT), TRIM21+/−, and TRIM21 KO mice (n = 4–6/group) were infected with HSV-1 (10^3^ PFU/cornea). At day 7 (**A**) or day 30 (**B**) post infection (PI), the mice were exsanguinated and the corneas were removed and assessed for opacity by optical density, using a plate reader, with tissue analyzed at 500-nm wavelength using a 30 × 30 matrix distributed over the cornea surface. Left panel is a representative cornea image from each mouse genotype captured at 500 nm. A summary of the results is shown in the right panel. Bars represent the mean + SEM. ** *p* < 0.01, *** *p* < 0.001 comparing the day 7 to day 30 pi timepoints, as determined by Kruskall–Wallis ANOVA and the Holm–Sidak post hoc *t*-test.

**Figure 4 viruses-14-00589-f004:**
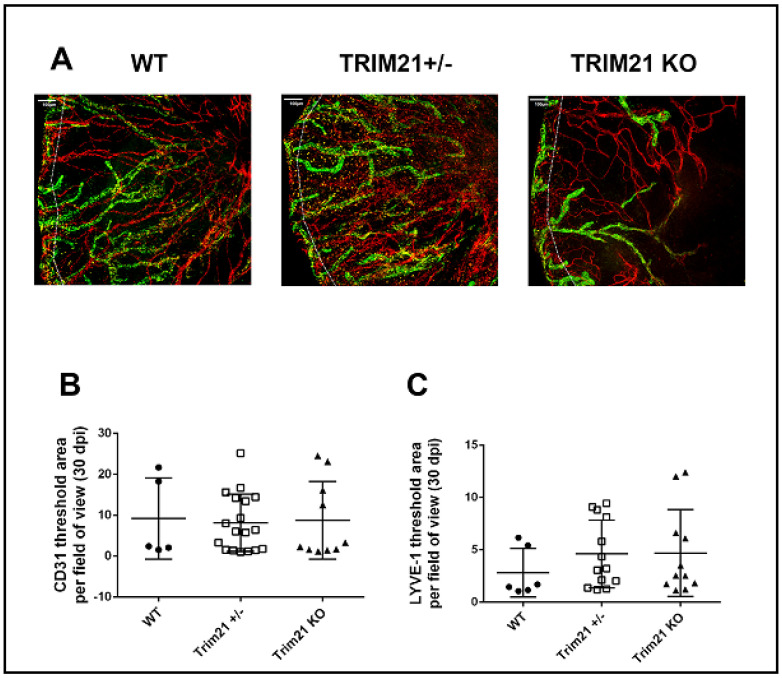
Corneal neovascularization is similar between WT, TRIM21+/− and TRIM21 KO mice in response to HSV-1 infection. Male and female C57BL/6 (WT), TRIM21+/−, and TRIM21 KO mice (n = 5–18/group) were infected with HSV-1 (10^3^ PFU/cornea). At 30 days post infection, the mice were exsanguinated, and the corneas were removed, processed, and stained for neovascularization measuring of lymphatic (green) and blood (red) vessel genesis into the central cornea, captured by confocal microscopy. (**A**) Representative images of corneas from WT, TRIM21+/−, and TRIM21 KO infected mice at day 30 pi. Dotted line depicts the limbus margins. Summary of the area occupied by (**B**) CD31+ blood and (**C**) LYVE-1+ lymphatic vessels for each group of mice. Horizontal bars represent the mean + SEM. The white scale bar = 100 µm.

**Figure 5 viruses-14-00589-f005:**
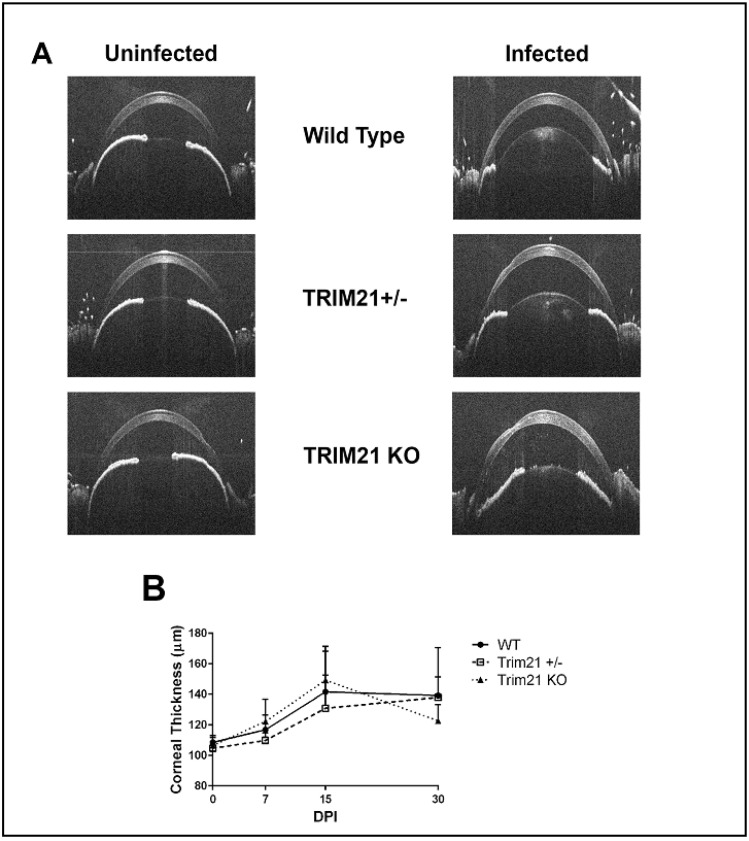
Cornea edema is similar between WT, TRIM21+/−, and TRIM21 KO mice in response to HSV-1 infection. Male and female C57BL/6 (WT), TRIM21+/−, and TRIM21 KO mice (n = 10–16/group/time point) were infected with HSV-1 (10^3^ PFU/cornea). At the indicated time point before (0) or after infection, the mice were sedated and the corneas were measured for thickness by spectral-domain optical coherence tomography. (**A**) Representative figures for each mouse genotype prior to (uninfected) and at day 15 post infection (infected). (**B**) Summary of corneal thickness for each mouse genotype at each time point, taken prior to and after infection. The points represent the mean + SD.

**Figure 6 viruses-14-00589-f006:**
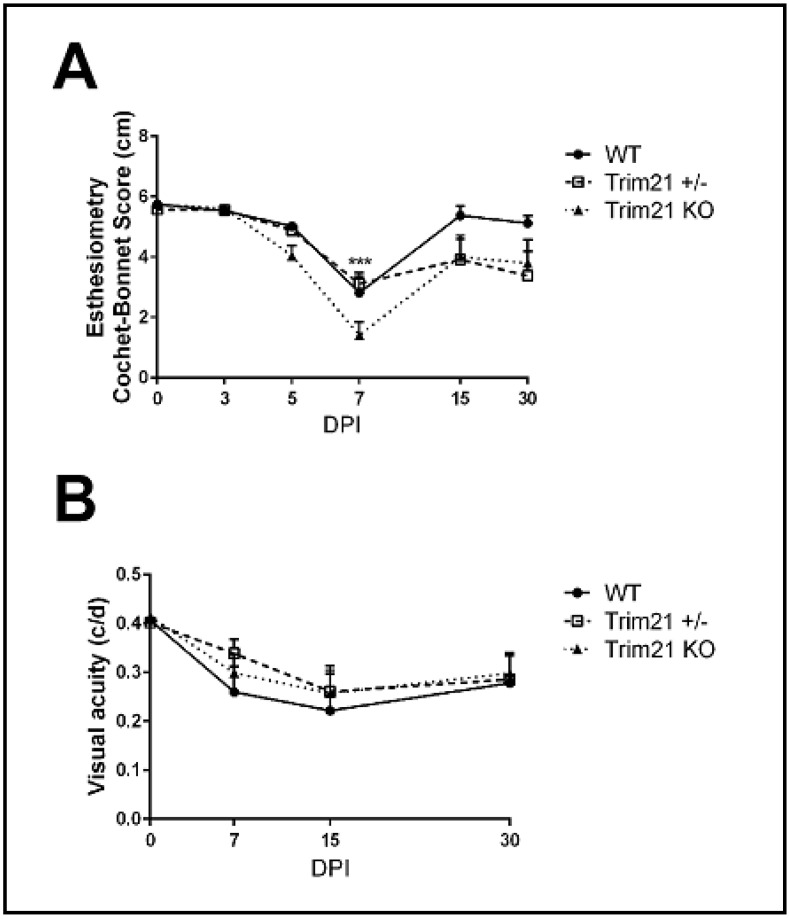
Performance of the visual axis drops in WT, TRIM21+/−, and TRIM21 KO mice following HSV-1 infection. Male and female C57BL/6 (WT), TRIM21+/− and TRIM21 KO mice (n = 5–15/group/time point) were infected with HSV-1 (10^3^ PFU/cornea). At the indicated time point, before (0) or after infection, the mice were evaluated for (**A**) mechanosensory function using a Cochet–Bonnet esthesiometer, to measure the blink reflex, or (**B**) assessed for visual acuity by optomotor kinetic tracking. All mice lost corneal sensation, with a maximum effect at 7 days post infection (DPI) and partial to complete recovery (in the case of WT mice) by 30 DPI (**A**). A 40% reduction in visual acuity was observed in all mouse groups, with a maximum effect at 15 DPI and partial recover by 30 DPI (**B**). Each point graphed represents the mean ± SEM. *** *p* < 0.001 comparing the 7 DPI to uninfected (0) time point, as determined by ANOVA and Tukey’s multiple comparison test.

**Figure 7 viruses-14-00589-f007:**
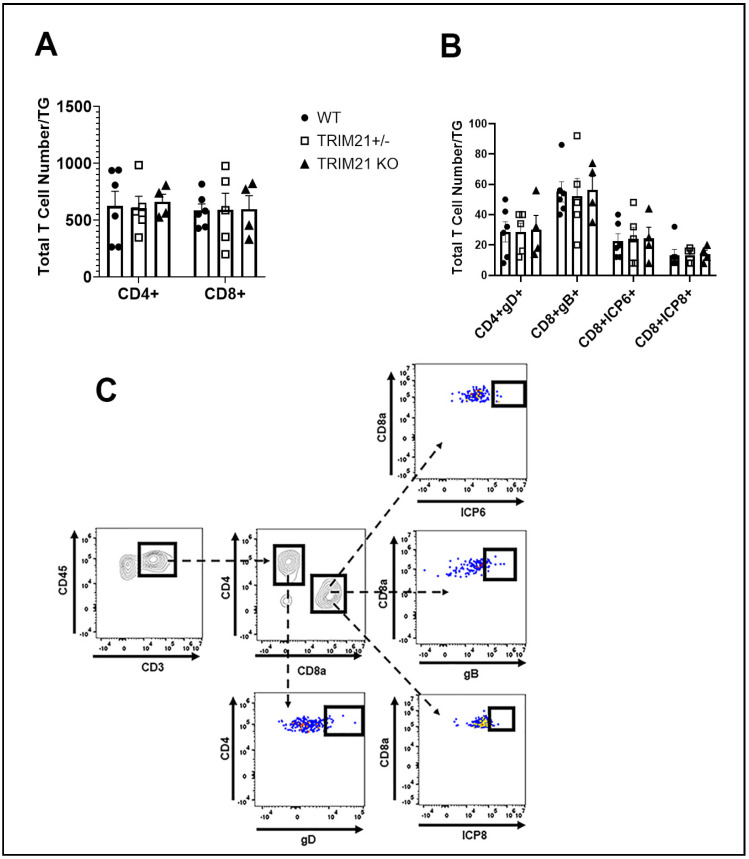
T cell infiltration in the TG of WT, TRIM21+/−, and TRIM21 KO mice at day 7 post infection is similar. Male and female C57BL/6 (WT), TRIM21+/−, and TRIM21 KO mice (n = 5–6/group) were infected with HSV-1 (10^3^ PFU/cornea). At day 7 pi, the mice were exsanguinated, and the TG were processed to single cell suspensions and stained for (**A**) total CD4+ and CD8+ T cells and (**B**) HSV-1 gD-specific CD4+ T cells and HSV-1 ICP6-, ICP8-, and gB-specific CD8+ T cells. The results are summarized, with the bars representing the mean + SEM. Uninfected mice possessed less than 25 CD4+ or CD8+ T cells/TG, with no HSV-1 tetramer-positive cells detected. (**C**) Gating strategy in the identification of cell populations acquired for panels A and B.

**Figure 8 viruses-14-00589-f008:**
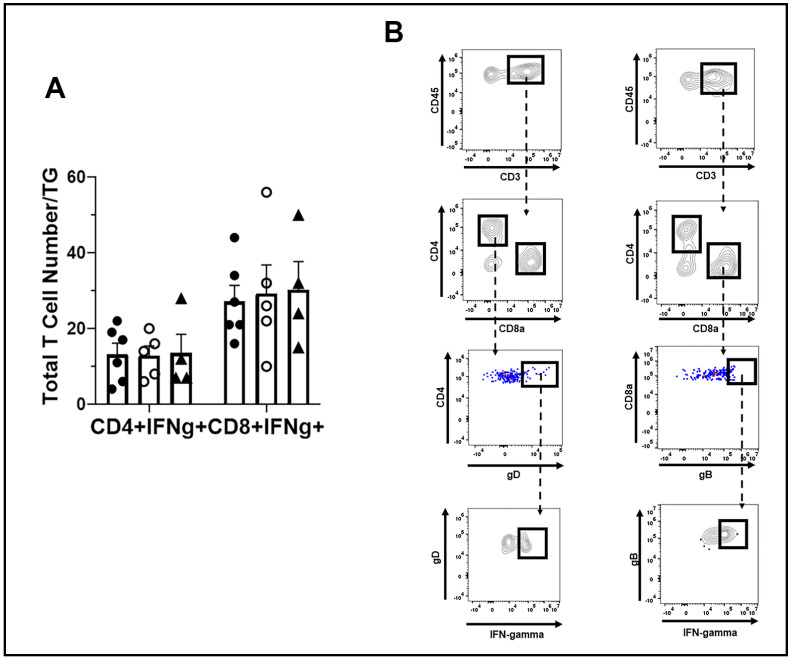
T cell function in the TG of WT, TRIM21+/− and TRIM21 KO mice, at day 7 post infection. Male and female C57BL/6 (WT), TRIM21+/−, and TRIM21 KO mice (n = 5–6/group) were infected with HSV-1 (10^3^ PFU/cornea). (**A**) At day 7 pi, the mice were exsanguinated, and the TG were processed to single cell suspensions and cultured for 6 h in the presence of PMA and ionomycin in the presence of brefeldin. Following these culture conditions, the cells were washed and stained for IFN-γ expressing HSV-1 gD-specific CD4+ and gB-specific CD8+ T cells and analyzed by flow cytometry. The results are summarized, with the bars representing the mean + SEM. No IFN-γ expressing cells were noted in unstimulated cultures. (**B**) Gating strategy in the identification of cell populations acquired for panel A.

**Figure 9 viruses-14-00589-f009:**
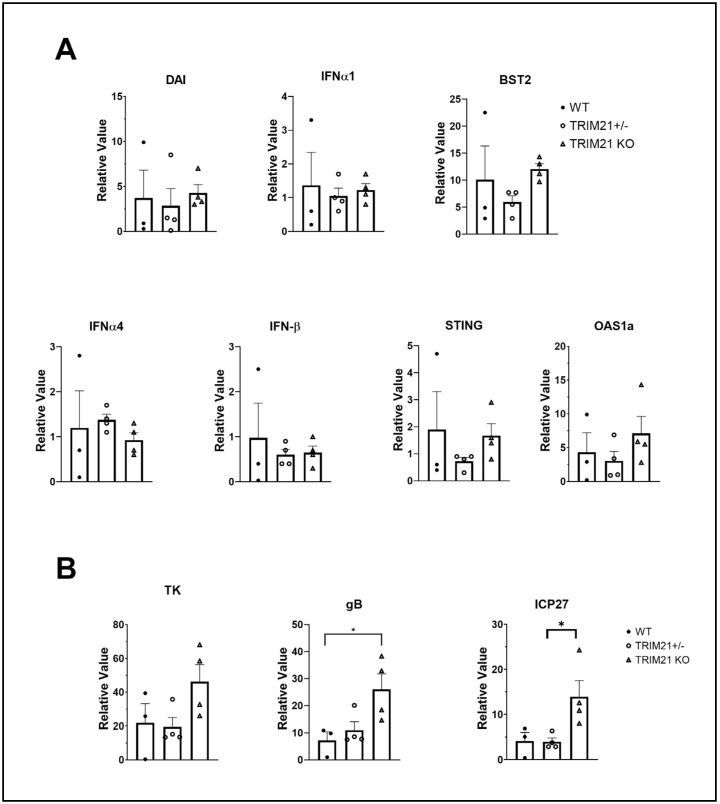
Elevation in HSV-1 lytic gene expression does not correlate with type I IFN gene expression. Female C57BL/6 (WT), TRIM21+/−, and TRIM21 KO mice (n = 4/group) were infected with HSV-1 (10^3^ PFU/cornea). At day 7 pi, the mice were exsanguinated, and the TG were removed and processed for select gene expression by real-time RT-PCR. (**A**) Relative values for Bst2 (tetherin), DNA-dependent activator of interferon regulatory factors (DAI), IFN-α1, IFN-α4, IFN-β, STING, and oligoadenylate synthetase (OAS)1a and (**B**) HSV-1 lytic genes infected cell protein (ICP)27, thymidine kinase (TK), and glycoprotein B (gB) were determined using the ∆Ct method, using uninfected genotypes as the controls to establish a baseline (relative value = 1). Bars represent the mean + SEM, * *p* < 0.05, comparing the indicated group as determined by Kruskall–Wallis ANOVA and the Holm–Sidak post hoc *t*-test.

**Table 1 viruses-14-00589-t001:** Primer Sequences for Targeted Genes.

	Forward	Reverse
IFNa4	5′-TTC TGC AAT GAC CTC CAT CA-3′	5′-GGC ACA GAG GCT GTG TTT CT-3′
mSTING	5′-CCT AGC CTC GCA CGA ACT TG-3′	5′-CGC ACA GCC TTC CAG TAG C-3′
DAI	5′-GGA AGA TCT ACC ACT CAC GTC-3′	5′-CCT TGT TGG CAG ATG ATG TTG-3′
Oas1a	5′-CTT TGA TGT CCT GGG TCA TGT-3′	5′-GCT CCG TGA AGC AGG TAG AG-3′
ICP27	5’- GCA TCC TTC GTG TTT GTC AT-3’	5’- ACC AAG GGT CGC GTA GTC-3’
TK	5′-ATA CCG ACG ATC TGC GAC CT-3′	5′-TTA TTG CCG TCA TAG CGC GG-3′
gB	5′-TCT GCA CCA TGA CCA AGT G-3′	5′-TGG TGA AGG TGG TGG ATA TG-3′

## Data Availability

The datasets generated and/or analyzed during the current study are available from the corresponding author upon reasonable request.

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
