# Peer review of "Tripartite-Motif 21 (TRIM21) Deficiency Results in a Modest Loss of Herpes Simplex Virus (HSV)-1 Surveillance in the Trigeminal Ganglia Following Cornea Infection"

_viruses, 2022, doi:10.3390/v14030589_

Round 1

Reviewer 1 Report

The manuscript is well written and explores the role of TRIM21 during HSV-1 corneal infections. The authors showed that TRIM21 knockout mice were no more susceptible to HSV-1 infection of the cornea. Higher viral titres were measured in the TG of TRIM21 knockout mice compared to WT, but no changes in IFN-gamma producing immune cells were observed. However there are several issues with the methodology that need to be addressed:

Line 102 - 103: minor typos, ensure numbers are superscript and the correct symbol for degrees is used

Line 106: missing the days pi that the mice were collected

Line 107: how was the cDNA generated? This sentence suggests that cDNA was generated using Trizol, which wouldn't be the case. Need to list commercial kit if one was used

Line 109: RT-PCR amplification conditions need to be specified. Need to also mention how PCR was performed. Commercial kit used, SYBR or probe based etc. It is unclear why for some genes forward and reverse primers are given, but for others a gene amplicon sequence was given. The primer pairs for each gene used in the study should be included in Table 1. How was the PCR normalised? I'm assuming GAPDH was used for normalisation as the sequence was mentioned in the methods, but this needs to be stated either in the methods or the corresponding figure legends. No mention of controls used. Were -RT and NTCs used to confirm specificity of primers?

Lines 135 - 136, 153 - 155, 161: details of antibodies are missing. Need to include the catalogue number for each antibody.

Line 159 and 163: details on fixation are lacking.

Line 170: specify what was in the antibody cocktail

Line 198 - 199: clarify the colours when referring to the confocal images. Currently the text says that both non-infected and infected cells are red. I'm assuming line 198 should be green instead of red?

Figure 1: scale bars for microscopy images are needed and should be included in the legend as well

Line 239: “the opacity increased as the infection progressed from day 7 to day 30 pi.” Is it possible to put the two graphs together and perform statistical analysis on this? Is there a significant increase in opacity between the two days or is this technically difficult to perform?

Figure 4: scale bars needed

Line 294: states that there was a significant loss in the blink response at day 15 pi, but the graph (6A) shows this loss at day 7

Figure 6: statistical test used for 6A is not stated. Also need to state what is plotted on the graph (mean + SEM?)

Figure 9: not all the data in this figure is referred to in the main text. This figure also needs to be reorganised. The graphs should be resized to all be the same size. The legend in the graph (indicating WT etc.) should be moved to the side rather than in the middle of the figure

Line 398 - 401 discuss that T cells may not reside in close proximity to HSV-1 infected cells in the TG. Can microscopy be performed to visualise T cells in the TG and to confirm that TRIM21 does not alter the localisation of these T cells?

Author Response

We would like to thank reviewer #1 for their helpful comments and suggestions.  Below is our response to each issue raised:

  1. Line 102 - 103: minor typos, ensure numbers are superscript and the correct symbol for degrees is used.

In the revised manuscript, we have corrected the error indicated by yellow highlight on line 103.

  1. Line 106: missing the days pi that the mice were collected.

In the revised manuscript, we have added “at the indicated days” indicated by yellow highlight on line 106 to describe when samples were collected as multiple time points were used and included for each figure shown.

  1. Line 107: how was the cDNA generated? This sentence suggests that cDNA was generated using Trizol, which wouldn't be the case. Need to list commercial kit if one was used.

In the revised manuscript we have changed the wording and included the cDNA synthesis kit as denoted by the yellow highlight on line 108.

  1. Line 109: RT-PCR amplification conditions need to be specified. Need to also mention how PCR was performed. Commercial kit used, SYBR or probe based etc. It is unclear why for some genes forward and reverse primers are given, but for others a gene amplicon sequence was given. The primer pairs for each gene used in the study should be included in Table 1. How was the PCR normalised? I'm assuming GAPDH was used for normalisation as the sequence was mentioned in the methods, but this needs to be stated either in the methods or the corresponding figure legends. No mention of controls used. Were -RT and NTCs used to confirm specificity of primers?

In the revised manuscript, we have made significant modifications.  Specifically, we stand corrected in the use of gene amplicons for select genes.  We used primer pairs for select gene targets.  The primer pairs were purchased from Bio-Rad and their sequences are proprietary information and unavailable but have been validated by the company. All other primer pair have previously been validated using melt curves as well as reactions in the absence of RT or template.  Additional information regarding method of amplification and normalization has been added to the revised manuscripts noted by yellow highlight from lines 111-125.  In addition, we noted the absence of three primer pairs targeting HSV-1 lytic genes including ICP27, TK, and gB.  Those primer pairs have been added to the table.

  1. Lines 135 - 136, 153 - 155, 161: details of antibodies are missing. Need to include the catalogue number for each antibody.

In the revised manuscript, from lines 168-173 we have added the clone and catalogue number for each antibody used in the study.  In addition, we have added the dye conjugate for our tetramers on lines 175 and 181 noted by yellow highlight in the revised manuscript.

  1. Line 159 and 163: details on fixation are lacking.

In the revised manuscript, we have included additional information of the fixation period from lines 183-186 noted by yellow highlight.

  1. Line 170: specify what was in the antibody cocktail.

In the revised manuscript, on lines 193-194 noted by yellow highlight the content of the antibody cocktail.

  1. Line 198 - 199: clarify the colours when referring to the confocal images. Currently the text says that both non-infected and infected cells are red. I'm assuming line 198 should be green instead of red?

In the revised manuscript, on lines 218 and 221 noted by yellow highlight we have clearly defined which cells express TRIM21 (red) and which are HSV-1-infected only(green).

  1. Figure 1: scale bars for microscopy images are needed and should be included in the legend as well.

In the revised manuscript, we have broadened the width of the scale bar (100 µm) which was apparently not visible in the original figure.

  1. Line 239: “the opacity increased as the infection progressed from day 7 to day 30 pi.” Is it possible to put the two graphs together and perform statistical analysis on this? Is there a significant increase in opacity between the two days or is this technically difficult to perform?

In the revised manuscript, we have incorporated the day 7 and day 30 pi data into a revised Fig. 3 and included the statistical analysis denoted in the figure legend noted by yellow highlight.

  1. Figure 4: scale bars needed.

In the revised manuscript, we have broadened the width of the scale bar (100 µm) in the revised figure.

  1. Line 294: states that there was a significant loss in the blink response at day 15 pi, but the graph (6A) shows this loss at day 7.

In the revised manuscript, we have corrected the error as noted by the reviewer.

  1. Figure 6: statistical test used for 6A is not stated. Also need to state what is plotted on the graph (mean + SEM?).

In the revised manuscript, noted by yellow highlight for lines 343-344 the following has been added, “Each point graphed represents the mean + SEM. ***p<.001 comparing the 7 DPI to uninfected (0) time point as determined by ANOVA and Tukey’s multiple comparison test.”

  1. Figure 9: not all the data in this figure is referred to in the main text. This figure also needs to be reorganised. The graphs should be resized to all be the same size. The legend in the graph (indicating WT etc.) should be moved to the side rather than in the middle of the figure.

In the revised manuscript, we have revised the figure as suggested by the reviewer.  In addition, as noted by yellow highlight we now include reference to STING and DAI on line 396.

  1. Line 398 - 401 discuss that T cells may not reside in close proximity to HSV-1 infected cells in the TG. Can microscopy be performed to visualise T cells in the TG and to confirm that TRIM21 does not alter the localisation of these T cells?

Thank you for the suggestion.  While it is possible to conduct such experiments, we currently do not possess TRIM21 KO mice.  We are down to one breeder in which the last litter provided only WT and TRIM21+/- mice.  It is likely she will again provide pups again of which some should be TRIM21 KO, the time to allow the animals to reach maturity for subsequent experimentation would be sometime in April at the earliest which would not meet the time line for the special issue.  Therefore, we have decided to forego experiments that might address this suggestion.

Reviewer 2 Report

This manuscript by Berube et al. looks at the role of host cell protein TRIM21 in the response of mice to HSV-1. Overall, the paper is clearly written and does a good job of laying out the rationale for looking at TRIM21 as a possible host defense factor against HSV-1. To investigate TRIM21, the authors compare the response of WT, TRIM21KO, and heterozygous mice to corneal infection by HSV-1. Most of the data are negative: there is no significant difference among the various groups of mice in virus replication in the eye, inflammation, or ocular pathology. Interestingly, however, the TRIM21 KO mice show ~10x enhanced replication in the trigeminal ganglion (TG) at day 7, and higher viral gene expression in TG at this same time point. The authors explore two reasonable explanations for this result (difference in the type I interferon or T cells responses), but find no difference among the groups in these parameters. Thus, in the end, we are left with a solid finding - TRIM21 suppresses HSV-1 replication in the mouse TG after ocular infection; however, the mechanism involved is unknown. Still, this is a step forward, and the science is very nicely done. I have only a few comments for the authors' consideration.

  1. The title of the paper seems to overstate the results (“Tripartite-motif 21 Deficiency Results in a Loss of HSV-1 Surveillance in the Trigeminal Ganglion….”). In the Abstract, the authors themselves call the effect of TRIM21 deficiency “modest” (line 23). It is ~10-fold – certainly a defect, but not necessarily a “loss of HSV-1 surveillance”, so the authors should consider modifying the title.

  1. What is the longer-term consequences of TRIM21KO in this system? Have the authors looked at a time point later than day 7 to see if the acute infection is ever resolved and if normal HSV-1 latency occurs? Of course, such experiments would be significantly more work, but would add a lot to the study.

  1. Figure 6A, which shows the results of the blink response test to assess eye function in the infected mice, is confusing. On line 293 the text reads “All groups of infected mice showed a significant loss in blink response at day 15 (Fig 6A)". Do the authors mean Day 7? What is being compared in the statistical test which shows high significance at Day 7- is this comparison KO mutant to WT? Also, why is the x axis not linear?

  1. Fig. 9 shows that the TGs from TRIM21KO-infected mice show enhanced viral gene expression. Do the authors know if this is due to more infected cells in these TGs, or is viral gene expression higher on a per cell basis? Since there are multiple cell types in TGs, is it possible that there is a difference in the target cells being infected?

Author Response

We would like to thank reviewer #2 for their helpful comments and suggestions.  Below is our response to each issue raised:

  1. The title of the paper seems to overstate the results (“Tripartite-motif 21 Deficiency Results in a Loss of HSV-1 Surveillance in the Trigeminal Ganglion….”). In the Abstract, the authors themselves call the effect of TRIM21 deficiency “modest” (line 23). It is ~10-fold – certainly a defect, but not necessarily a “loss of HSV-1 surveillance”, so the authors should consider modifying the title.

In the revised manuscript, we have added the word “Modest” before “Loss” in the title.

  1. What is the longer-term consequences of TRIM21KO in this system? Have the authors looked at a time point later than day 7 to see if the acute infection is ever resolved and if normal HSV-1 latency occurs? Of course, such experiments would be significantly more work, but would add a lot to the study.

The concept behind the paper (in part, based on previously published work) was to investigate (i) the acute infection both in terms of virus replication, virus spread, and the host immune response.  In addition, the longer term consequences of virus infection on the visual axis was added as this is perhaps the most important clinical aspect of the infection.  Characterizing the visual axis function and pathology found no apparent difference comparing WT to TRIM21 KO mice.  Therefore, we did not feel it would benefit the study to conduct additional experiments since our clinical readouts did not show differences.  Although not included, we did measure neutralizing antibody titers from sera obtained from animals at day 30 pi.  However, the results did not identify any differences in either titer or antibody isotype associated with antigen recognition comparing WT to TRIM21 KO mice.

  1.  Figure 6A, which shows the results of the blink response test to assess eye function in the infected mice, is confusing. On line 293 the text reads “All groups of infected mice showed a significant loss in blink response at day 15 (Fig 6A)". Do the authors mean Day 7? What is being compared in the statistical test which shows high significance at Day 7- is this comparison KO mutant to WT? Also, why is the x axis not linear?

In the revised manuscript, we have changed the noted difference to day 7 from day 15.  We apologize for the error.  We have added the following noted by yellow highlight on lines 343-344 to the figure legend of Fig. 6A clarifying comparisons and statistical test used, “Each point graphed represents the mean + SEM. ***p<.001 comparing the 7 DPI to uninfected (0) time point as determined by ANOVA and Tukey’s multiple comparison test.”

  1. 9 shows that the TGs from TRIM21KO-infected mice show enhanced viral gene expression. Do the authors know if this is due to more infected cells in these TGs, or is viral gene expression higher on a per cell basis? Since there are multiple cell types in TGs, is it possible that there is a difference in the target cells being infected?

As indicated in the manuscript, the data reflects relative value for targeted lytic gene expression for each genotype using the uninfected control to establish the baseline (relative value = 1) now included on lines 409-410 noted in yellow highlight in the revised manuscript.  There are many possibilities that may have resulted in changes in virus replication comparing the TRIM21 KO to WT mice some of which are indicated by the reviewer.  We suggested in the original and revised manuscript that apoptotic or necrotic events could contribute to the susceptibility to virus replication comparing the WT to TRIM21 KO mice as such has been reported by other groups employing different viruses as well as HSV-1 noted in our discussion.

Round 2

Reviewer 1 Report

None